



# Soil moisture observation in a forested headwater catchment: combining a dense cosmic-ray neutron sensor network with roving and hydrogravimetry at the TERENO site Wüstebach

Maik Heistermann[1], Heye Bogena[2], Till Francke[1], Andreas Güntner[3,1], Jannis Jakobi[2], Daniel Rasche[3,1], Martin Schrön[4], Veronika Döpper[5], Benjamin Fersch[6], Jannis Groh[2,7], Amol Patil[8], Thomas Pütz[2], Marvin Reich[3], Steffen Zacharias[4], Carmen Zengerle[4], and Sascha Oswald[1]

[1]Institute of Environmental Science and Geography, University of Potsdam, Karl-Liebknecht-Straße 24–25, 14476 Potsdam, Germany
[2]Agrosphere IBG-3, Forschungszentrum Jülich GmbH, Leo-Brandt-Straße, 52425 Jülich, Germany
[3]GFZ German Research Centre for Geosciences, Section Hydrology, Telegrafenberg, 14473 Potsdam, Germany
[4]UFZ – Helmholtz Centre for Environmental Research GmbH, Dep. Monitoring and Exploration Technologies, Permoserstr. 15, 04318, Leipzig, Germany
[5]Technical University of Berlin, Geoinformation for Environmental Planning Lab, Straße des 17. Juni 135, 10623 Berlin, Germany
[6]Karlsruhe Institute of Technology, Campus Alpin (IMK-IFU), Kreuzeckbahnstraße 19, 82467 Garmisch-Partenkirchen, Germany
[7]Leibniz Centre for Agricultural Landscape Research (ZALF), Eberswalder Str. 84, 15374 Müncheberg, Germany
[8]Institute of Geography, University of Augsburg, Alter Postweg 118, 86159 Augsburg, Germany

**Correspondence:** Maik Heistermann (maik.heistermann@uni-potsdam.de)

**Abstract.**

Cosmic Ray Neutron Sensing (CRNS) has become an effective method to measure soil moisture at a horizontal scale of hundreds of meters and a depth of decimeters. Recent studies proposed to operate CRNS in a network with overlapping footprints in order to cover root-zone water dynamics at the small catchment scale, and, at the same time, to represent spatial

5 heterogeneity. In a joint field campaign from September to November 2020 (JFC-2020), five German research institutions deployed 15 CRNS sensors in the $0.4\,km^2$ Wüstebach catchment (Eifel mountains, Germany). The catchment is dominantly forested (but includes a substantial fraction of open vegetation), and features a topographically distinct watershed. In addition to the dense CRNS coverage, the campaign featured a unique combination of additional instruments and techniques: hydro-gravimetry (to detect water storage dynamics also below the root zone); ground-based and, for the first time, airborne

10 CRNS roving; an extensive wireless soil sensor network, supplemented by manual measurements; and six weighable lysimeters. Together with comprehensive data from the long-term local research infrastructure, the published dataset (available at https://doi.org/10.23728/b2share.afb20a34a6ac429ca6b759238d842765) will be a valuable asset in various research contexts: to advance the retrieval of landscape water storage from CRNS, wireless soil sensor networks, or hydrogravimetry; to identify scale-specific combinations of sensors and methods to represent soil moisture variability; to improve the understanding

15 and simulation of land-atmosphere exchange as well as hydrological and hydrogeological processes at the hill-slope and the catchment scale; and to support the retrieval soil water content from airborne and spaceborne remote sensing platforms.





# 1 Introduction

## 1.1 The estimation of soil water content by cosmic-ray neutron sensing

The spatial representativeness of conventional point-based soil moisture measurements is often limited, specifically in case the small measurement support comes up against high small-scale variability (Blöschl and Grayson, 2000). In-situ point methods often use electromagnetic (EM) approaches, e.g. frequency domain reflectometry (FDR), time domain transmission (TDT), time domain reflectometry (TDR), or capacitance (Kojima et al., 2016) and impedance sensors (Wilson et al., 2020). Alternatively, remote sensing techniques can deliver area-integrated measurements, however, they are often confronted with low overpass frequencies, large spatial footprints, as well as shallow penetration depths, making root-zone soil moisture retrieval difficult (Peng et al., 2021). In that context, the presence of dense vegetation layers, such as forests, remains a major source of uncertainty (Li et al., 2021).

Over the past decade, various techniques have emerged to address such issues of vertical and horizontal representativeness, and to close the scale-gap between point measurements and large-scale soil moisture retrievals (Fig. 1). Of these techniques, cosmic-ray neutron sensing (CRNS) has attracted particular attention, and various application scenarios have been developed that target different spatial and temporal scales.

A **single, stationary CRNS sensor** can be used to obtain a volume-integrated measurement of soil moisture (Zreda et al., 2008) which is representative for a horizontal radius of 100-150 m, a vertical depth of 20-50 cm. Depending on the detector sensitivity, Schrön et al. (2018) found the effective temporal resolution of such measurements to range from three to twelve hours. The sensor measures the ambient above-ground density of epithermal neutrons (at energies of about $1$–$10^5$ eV) which is inversely related to the abundance of hydrogen and, consequently, of soil moisture (Köhli et al., 2020). Water in the soils can be quantified from this epithermal neutron intensity by conversion functions (e.g., Desilets et al., 2010; Köhli et al., 2020), which usually require the calibration of a detector-specific scaling parameter (e.g., $N_0$) on independent soil moisture measurements in the footprint of a neutron detector (see Schrön et al., 2017, for a recent synthesis).

Soon enough, a **mobile CRNS sensor ("CRNS roving")** was established as a means to detect patterns of soil water content along transects across the landscape (see e.g. Schrön et al., 2018). To that end, the sensor is moved within the area of interest (e.g. by car, or, as recently suggested, by train, see Schrön et al.). This way, the technique could cover up to a few hundreds of square kilometers within a day. The corresponding spatial representativeness of the measurements depends on the accessible road network and travel speed, while road material itself may introduce bias (Schrön et al., 2018). In contrast to stationary sensors, however, CRNS roving can explore the spatial distribution of soil moisture as a single snapshot in time, which is a an advantage when it comes to spatial coverage of a region.

Recently, a **dense stationary CRNS network** was explored as a means to resolve, continuously in time, the spatial heterogeneity within and between CRNS footprints (Heistermann et al., 2021b). The first realization of such a dense CRNS network had been implemented in a joint field campaign from May to July 2019 during which 24 CRNS sensors were operated in the Rott headwater catchment, a pre-alpine area of 1 km² in southern Germany. The corresponding CRNS data were published by





Fersch et al. (2020), together with comprehensive additional data required to correct, calibrate, and validate CRNS-based soil moisture retrievals.

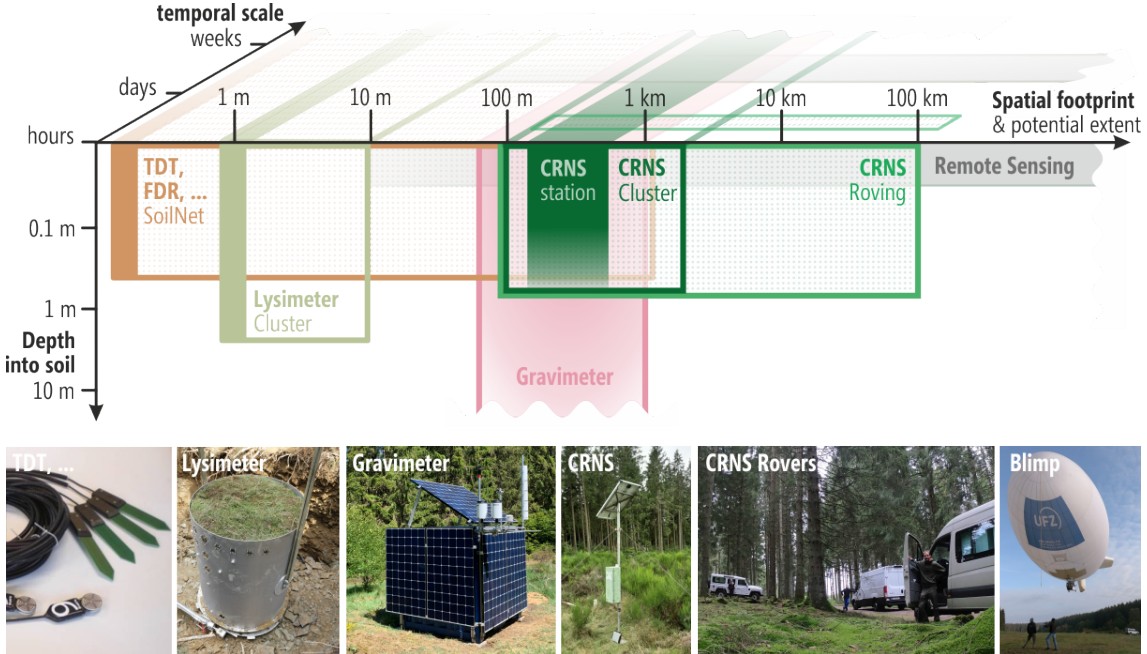

**Figure 1.** Scales of soil moisture observations in the JFC-2020. The figure aims to convey a general idea of how the various instrumental platforms differ with regard to their representativeness along horizontal, vertical and temporal dimensions. Solid areas indicate the footprint (support, resolution) of an individual sensor while the frames show the minimum resolution and maximum spatial extent of typical sensor applications (e.g. in terms of dense networks or measurements along transects). Although not directly part of the JFC-2020, remote sensing products such as Sentinel 1, SMAP (Soil Moisture Active Passive), or ASCAT (Advanced Scatterometer) are shown for the sake of comparability. Please see subsections 3.3-3.9 for details on the specific measurement methods.

## 1.2 The Cosmic Sense Joint Field Campaign in 2020

From September to November 2020, the research unit *Cosmic Sense*, funded by the German Research Foundation (DFG), ventured a second massive effort to explore the potential of dense stationary CRNS networks for monitoring spatial and
temporal soil moisture dynamics at the catchment scale. This effort will be referred to as the "Joint Field Campaign 2020 (JFC-2020)" (while we refer to the first JFC in 2019, as published by Fersch et al. (2020), as *JFC-2019*). A graphical overview of the different instruments and their observational scales is provided in Fig. 1.

The JFC-2020 featured 15 stationary CRNS sensors in an area of $0.39\,\text{km}^2$, the Wüstebach headwater catchment in the Eifel, a low mountain range in western Germany, and was motivated most importantly by the aim to





- **maximize coverage**: monitor water storage in the root zone of an entire catchment, and to possibly relate storage changes to the catchment's runoff response by means of hydrological modelling;

- **maximize resolution**: use the overlap between CRNS sensor footprints to establish a spatio-temporal representation of soil moisture patterns at potentially higher resolutions than the footprint of a single sensor, e.g. by means of interpolation.

As in the earlier JFC in 2019, the campaign was integrated with long-term observational infrastructure provided by TERENO (Zacharias et al., 2011), and, again, the dense CRNS network was complemented by CRNS roving, manual ground truth measurements, as well as biomass and soil mapping - all in a study catchment that is characterized by pronounced soil moisture heterogeneity in space.

However, the JFC-2020 exhibited a number of important features which were new or essentially different in comparison to the JFC-2019, and which, in their entirety, make the JFC-2020 a unique contribution. These differences particularly emerge with regard to the properties of the study area and additional instrumental features, as elaborated in the following list:

- **Different season, different processes**: The JFC-2019 observational period from late spring to midsummer started out from fully saturated conditions, followed by two months of marked drying. For the JFC-2020, we chose, instead, to monitor the re-wetting process from late summer to late autumn which we expected to follow different vertical soil moisture dynamics (as compared to drying), with likely implications for the dynamics of the CRNS penetration depth;

- **Addressing a challenging land cover**: In contrast to the JFC-2019, we chose a dominantly forested study area for the JFC-2020. With regard to CRNS, forests are a more challenging environment due to a lower epithermal neutron intensity and a typically large spatial heterogeneity of vegetation biomass and soil moisture;

- **More pronounced watershed boundaries**: In comparison to JFC-2019, the study area is more topographically structured, hence the above-ground watershed is more pronounced. This should allow for a better closure of the catchment's water balance, and thus an increased potential for hydrological applications and studies that relate the root-zone water balance to observed discharge;

- **Taking the CRNS rover to another dimension:** For the first time, an experiment with an airborne CRNS roving platform was conducted that was mounted on an airship and repeatedly crossed the study area at altitudes between 10 an 170 m above the ground. We consider airborne CRNS roving as an opportunity to reduce the influence of the near field (e.g. by roads), to decrease the dependency on transport networks on the ground, and to reshape the horizontal CRNS footprint.

- **Beyond near-surface soil moisture with hydrogravimetry:** The JFC-2020 featured a gravimeter at the eastern hill-top of the catchment. That way, we expected to capture the dynamics of water storage below the vertical penetration depth of the CRNS technology, i.e., in the deeper unsaturated zone, including the entire root zone. In combination with CRNS, we hypothesize that gravimetry could provide vital information to detect vertical and horizontal water exchange, and hence to help closing the catchment's water balance;





- **Higher stationary CRNS density:** With a catchment size of about $0.4\,\text{km}^2$ (as compared to $1\,\text{km}^2$ in 2019) and no access restrictions, the JFC-2020 could provide full CRNS coverage of the study area and, at the same time, operate with a higher overlap of CRNS footprints. That way, we provide strong support for both research aims: coverage *and* resolution;

- **Unique ground truth coverage in space and time**: The TERENO infrastructure in the Wüstebach catchment features a local wireless soil sensor network (WSN/SoilNet, 101 active nodes during the campaign), as well as one location with six weighable lysimeters. These data are part of the published data set, and provide substantial ground truth coverage in both space and time. This is a huge advantage as compared to JFC-2019 where the WSN only covered about 10 % of the study area;

- **Learning from the past**: Learning from technical issues in sensor maintenance during JFC-2019 (specifically regarding power supply and real-time power monitoring) lead to a dramatic decrease in data gaps for the stationary CRNS network, which improves our ability to provide continuous sensor coverage over the whole campaign period.

### 1.3 Structure of this paper

In this paper, we present the data set obtained in the JFC-2020. It is available via EUDAT at (Heistermann et al., 2021a). Section 2 introduces the study area. In section 3, we document the different subsets of data, the data collection, and, if applicable, the involved data processing. In section 4, we highlight data that is expected to be relevant for the analysis of the published data, but which is available from other providers and hence not included in this publication. Section 5 exemplarily illustrates properties of the obtained data with regard to the spatio-temporal representation of soil moisture and water storage. Section 6 concludes by highlighting perspectives for prospective research with the presented data set.

### 2 Study site

Different criteria were considered for the selection of the study site, some of which have already been touched upon in section 1.2. Most importantly, the study area was required

- to be predominantly forested: as forested areas turned out to be challenging environments for CRNS-based soil moisture retrieval (Bogena et al., 2013), we intended to demonstrate the feasibility of the dense CRNS network concept also in a dominantly forested area;

- to have a well-defined watershed: This should help users of the data set to explore the added value of catchment-scale soil moisture retrieval for hydrological (namely rainfall-runoff) modelling;

- to contain areas with sufficient distance to the groundwater table in order for the hydrogravimetric measurements to be sufficiently sensitive to changes in water content dynamics in the unsaturated zone;



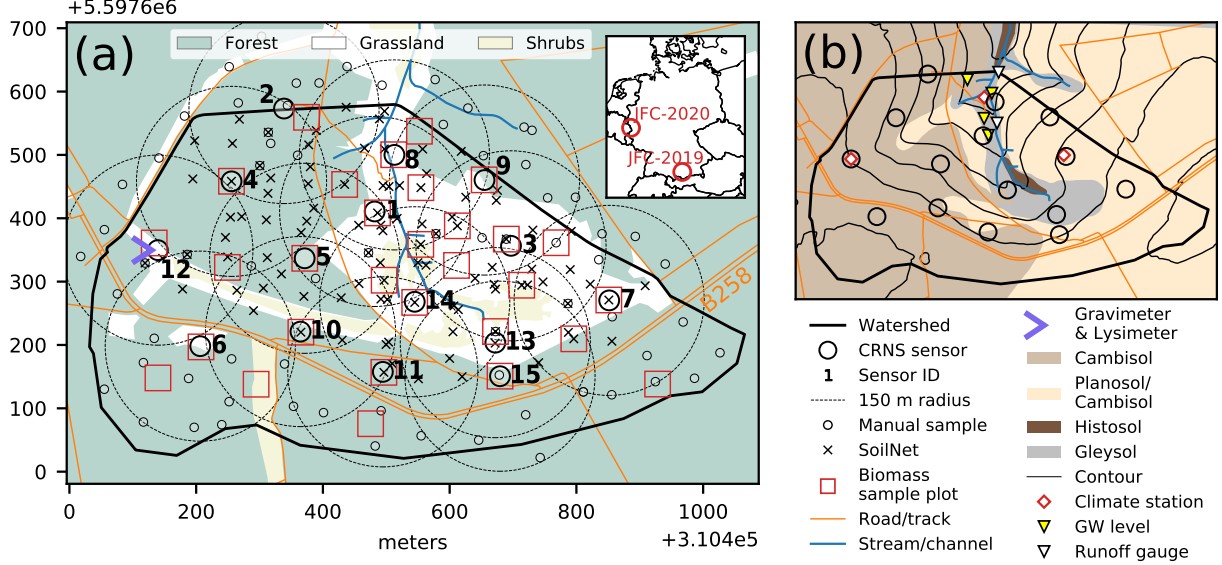

**Figure 2.** The Wüstebach headwater catchment, about 30 km south-west to the city of Aachen, Nordrhein-Westfalen, Germany. (a) locations of the CRNS sensors, together with a 150 m radius (approximate footprint radius for intermediate soil moisture, Schrön et al., 2017), the SoilNet nodes active during the campaign, the manual soil sampling (October 19, 2020), the biomass sample plots, the gravimeter and the lysimeters; (b) soil types according to FAO classification from (Richter, 2007), contour lines, and locations of TERENO climate stations, runoff gauges and groundwater monitoring wells; OSM layers were used to represent roads, landuse, and waterways (© OpenStreetMap contributors, 2021. Distributed under the Open Data Commons Open Database License (ODbL) v1.0.).

- to feature long-term, hydrological and hydro-meteorological observation infrastructure, specifically with regard to continuous ground measurements of soil moisture;

- to have good accessibility by foot and car for installation, maintenance, CRNS roving as well as manual measurements;

- to be sufficiently close to at least one of the participating research institutions to allow for instrument maintenance. This practical aspect was particularly critical since we targeted a forested catchment in autumn with the consequence that local insolation was not sufficient for solar power supply at all measurement locations, and batteries had to be regularly changed.

Based on these criteria, we selected the Wüstebach catchment (Fig. 2), a 0.39 km$^2$ headwater catchment of the Rur river (Eifel mountains in western Germany, close to the Belgian border) which is part of the Terrestrial Environmental Observa-
tory (TERENO-Rur, Bogena et al., 2018) of the Helmholtz association.

The Wüstebach experimental catchment was extensively investigated and documented by many studies: to name just a few, Bogena et al. (2018) provided a general overview of climate, (hydro-)geology, land use, and core instrumentation; Gottselig et al. (2017a) published a dataset on physical and chemical soil properties; Bogena et al. (2013) and Baatz et al. (2015)





investigated the estimation of soil moisture from three permanently installed CRNS sensors; while Bogena et al. (2010) and

Graf et al. (2014) examined the potential of the wireless soil sensor network in the area to represent the distribution of soil moisture in space and time. Due to the comprehensive documentation of the Wüstebach catchment in the scientific literature, we will keep the description of the study area very brief, quoting the overview on page 9 of Bogena et al. (2018):

> "The altitude [of the catchment ranges from] 595 to 628 m asl [...]. The geology is dominated by Devonian shales [which are] covered by a 1-2 m deep periglacial solifluction layer in which Cambisols and Planosols have devel-
oped on the hillslopes while Gleysols and Histosols dominate under the influence of groundwater in the valley. The main soil texture is silty clay loam; mean annual precipitation is 1220 mm [...], and the main vegetation is Norway spruce [Picea abies (L.) H. Karst.] planted in 1946. During the late summer of 2013, trees were almost completely removed in an area of 9 ha near the main Wüstebach stream to initiate the regeneration of near-natural beech (Fagus sylvatica L.) forest."

## 3  Methods and data

### 3.1  Overview

This section describes the data obtained during the JFC-2020 campaign, and, if applicable, the corresponding data processing methods. The prime motivation of the JFC-2020 was the application of a dense network of 15 CRNS sensors in a catchment of 0.39 km$^2$ (section 3.3). Additional measurements were carried out to study and evaluate the soil moisture retrieval from
observed neutron counts, and to put it into context with other observations methods: reference CRNS measurements were collected to standardize neutron counts rates from different sensors (section 3.4); CRNS roving was conducted on ground-based (section 3.5) and airborne (section 3.6) platforms; a gravimeter recorded temporal variations of the local gravity field (section 3.7); soil properties and soil moisture measurements for calibration and validation purposes were taken at selected locations and at different depths (section 3.8); weight and percolation were recorded by six weighable lysimeters (section 3.9);
and, finally, above-ground biomass was mapped to quantify the corresponding hydrogen pools which may influence the CRNS signal (section 3.10).

### 3.2  Data formats

The overall data set is organized along instruments and observed variables (section 7), and each subset of data is documented in a dedicated meta-data file in "json" format. As in Fersch et al. (2020), the presented data largely consists of time series
that were obtained at well-defined locations (e.g., neutron counts, or soil water content). For such data, we implemented a transparent and simple data model based on text tables (character separated values, csv). For sensor networks (e.g. CRNS, SoilNet), an overview table provides a unique identifier (ID) for each sensor unit, as well as its location (longitude, latitude, in WGS 84 reference system), and, if applicable, additional sensor attributes. The observed time series are then provided in additional csv-files in which the first column contains date and time (in UTC, ISO 8601 format). Any other columns represent



measured or derived variables. Exceptions from this data model (e.g., for roving, vegetation, manual soil measurements) are described in the subsections of this paper, and documented by json-files with meta-data. For vector geodata, the format of ESRI shapefiles was used. Further details on the data repository are given in section 7.

### 3.3   Stationary CRNS data

In the core campaign period from September 1 to November 1, 15 stationary CRNS sensors were operated. Three CRNS
sensors (IDs 1-3) were part of the permanent TERENO instrumentation. Tab. 1 provides an overview of the stationary CRNS sensors. Out of the 15 sensors, 13 had been manufactured by Hydroinnova LLC (Albuquerque, NM, USA), and two by Lab-C LLC (Tucson, AZ, USA). All instruments were based on neutron-sensitive detector gases, such as $^3$He gas (CRS-1000, CRS-2000) or $^{10}$BF$_3$ enriched gas (CRS-1000-B, CRS-2000-B, B-E1-4).

The neutron detection chamber in each sensor unit was enclosed with a material (moderator) to "thermalize" (i.e., slow down)
the ambient epithermal neutrons prior to detection (Zreda et al., 2012; Köhli et al., 2018; Schrön et al., 2018). An additional detection mode (a "bare counter" without such a moderator) was active in four sensors to count thermal neutrons instead (see Tab. 1), as several studies had suggested that the ratio between thermal and epithermal neutron intensities could be used to distinguish hydrogen from soil moisture and vegetation (Tian et al., 2016; Jakobi et al., 2018).

Furthermore, the CRNS sensor units recorded relative humidity, air temperature, and barometric pressure. These variables
are necessary to later take into account atmospheric effects on the observed neutron intensities; further details are available in the attribute table of the stationary CRNS data subset. The measurement interval for the CRNS sensors was set to 20 minutes for most sensors (except sensors 1 and 3 with 60 minutes, and sensor 2 with 15 minutes). Data gaps between September 1 and November 1 are negligible, except for sensor 3 which misses data from September 20, 17:00, to 22, 9:00 UTC.

The locations of the CRNS sensors are shown in Fig. 2. The overarching scientific aim was to achieve, on the one hand,
a high CRNS coverage of the catchment, and, on the other hand, to have a strong overlap between CRNS footprints. Both requirements could be better met in JFC-2020 as compared to JFC-2019, due to the lack of access restrictions, and the higher number of CRNS sensors per area. Additional scientific and practical constraints for sensor placements were:

- to place, if possible, each CRNS sensor close to a SoilNet node (section 3.8.1) in order to monitor the vertical soil moisture distribution in close proximity (see Tab. 1, column SoilNet ID). Sensor #6 was located outside the SoilNet
coverage, but was equipped with a profile probe instead (section 3.8.3).

- to balance the coverage between forested and deforested areas, as well as between the soils in the wetter valley bottom (Gleysols, Histosols) and those that have developed on the hill slopes (Cambisols and Planosols).

- the previous criterion is in line with the requirement to include sites with a close groundwater table in the valley and a distant groundwater table on the hill tops. For the latter, one CRNS location (#12) had to coincide with the placement of
the gravimeter for the sake of comparability;

**Table 1.** Properties of CRNS sensors used in the JFC-2020, including manufacturer, model, and detector gas; the availability of detector tubes for epithermal neutrons (moderated - mod) and "bare" tubes for the detection of thermal neutrons; the predominant vegetation in the sensor footprint; the ID of the nearest SoilNet node, if available; the ratio of the sensor's raw counts of neutrons to the counts of a calibrator sensor (#16), referred to as sensitivity factor.

| ID | Manufacturer | Sensor model | Detector gas | Tubes | Dominant land cover | SoilNet ID | Sensitivity |
|---|---|---|---|---|---|---|---|
| 1 | Hydroinnova | CRS 1000 | $^3$He | mod & bare | mostly clear-cut | 75 | 0.452* |
| 2 | Hydroinnova | CRS 2000-B | $^{10}$BF$_3$ | mod & bare | forest | 52 | 1.147* |
| 3 | Hydroinnova | CRS 1000 | $^3$He | mod & bare | clear-cut | 95 | 0.452* |
| 4 | Hydroinnova | CRS 2000-B | $^{10}$BF$_3$ | mod | forest | 13 | 1.147* |
| 5 | Hydroinnova | CRS 2000-B | $^{10}$BF$_3$ | mod | forest | 86 | 1.147* |
| 6 | Hydroinnova | CRS 2000-B | $^{10}$BF$_3$ | mod | forest | – | 1.147* |
| 7 | Hydroinnova | 2 x CRS 1000 | $^3$He | mod | clear-cut, forest | 118 | 0.904 |
| 8 | Lab-C | B-E1-4 | $^{10}$BF$_3$ | mod | clear-cut, forest | 9 | 2.030 |
| 9 | Lab-C | B-E1-4 | $^{10}$BF$_3$ | mod & bare | forest, clear-cut | 20 | 2.473 |
| 10 | Hydroinnova | 2 x CRS 1000-B | $^{10}$BF$_3$ | mod | forest | 122 | 1.255 |
| 11 | Hydroinnova | 2 x CRS 1000-B | $^{10}$BF$_3$ | mod | forest | 142 | 1.400 |
| 12 | Hydroinnova | CRS 2000-B | $^{10}$BF$_3$ | mod | forest | 100 | 1.276 |
| 13 | Hydroinnova | CRS 2000-B | $^{10}$BF$_3$ | mod | clear-cut, forest | 133 | 1.152 |
| 14 | Hydroinnova | CRS 2000-B | $^{10}$BF$_3$ | mod | clear-cut | 49 | 1.102 |
| 15 | Hydroinnova | CRS 2000-B | $^{10}$BF$_3$ | mod | forest, clear-cut | 133 | 1.191 |
| 16 | Hydroinnova | Calibrator | $^3$He | mod | not applicable | – | 1.000 |

[*] Direct measurement unavailable, sensitivity was estimated from the average sensitivity of sensors of the same model (Fersch et al., 2020).

- to seek for locations with increased insolation in a mainly forested area (e.g. small forest glades, forest edges, and the like) to ensure solar power supply;

- to keep a distance of at least 15 m to roads and other structures which could introduce bias to the CRNS measurements (Schrön et al., 2018).

## 200  3.4  Standardisation of neutron count rates

In order to achieve comparability of the observed neutron counts rates across different sensor types and models, we used a mobile "calibrator" sensor (Hydroinnova, #16 in Tab. 1) as a common reference standard. We collocated the calibrator with selected stationary sensors for at least 24 hours. The ratio between the average count rates of the two instruments is defined as the sensitivity factor and serves as standardization of the different sensors against the calibrator level. Sensors which could
not be collocated with the calibrator received previously-obtained sensitivity factors from earlier campaigns, e.g., during JFC-2019. This approach is adequate as the sensitivity can be assumed to be effectively time-invariant (Heistermann et al., 2021b). The resulting sensitivity factors are included in Tab. 1.



## 3.5 Roving CRNS

Mobile CRNS measurements were conducted to acquire snapshots of the spatial soil moisture distribution in the catchment.
On Oct 19 and 20, CRNS roving was performed with three different cars, twice with a hot-air blimp (manned maneuverable hot-air balloon, see section 3.6), and once by two persons carrying a CRNS rover-unit to access otherwise inaccessible areas. Three variable detector systems were used to monitor different energy levels:

- **HI1:** The UFZ Hydroinnova rover is a moderated CRNS unit (Hydroinnova LLC, Albuquerque, USA) based on $^3$He gas (see Schrön et al., 2018; Fersch et al., 2020, for details) and mounted in a car (Land Rover Defender). In order to reduce potential local effects beneath the car, the detector has been equipped with an additional polyethylene shield of 5 cm thickness at the bottom. The detector has been used in two experiments on a blimp, two times in a car, once carried by hand, and once accompanied the rover system SN8 (see below) in a van. In addition, geotagged camera images have been taken at every 20 seconds using three cameras on board of the off-road car, at a total view angle of 270 degree (left, right, back). One camera has been used also aboard the blimp. The images can be useful to interpret the measurements, and to characterize possible influences of road or vegetation.

- **HI9S:** The FZJ Hydroinnova system on board of Van A (Mercedes Sprinter) (Jakobi et al., 2020) allowed to simultaneously measure epithermal, thermal, and thermal-shielded signals in both, horizontal and vertical modes (Köhli et al., 2018). Each of the nine neutron detector units holds four tubes filled with $^{10}$BF$_3$. Five of the active detectors counted epithermal neutrons, two in horizontal and three in vertical orientation. One vertical epithermal unit was additionally shielded with gadolinium to exclude thermal contamination. In order to further explore the potential of the ratio between thermal and epithermal neutrons intensities (see, e.g., Tian et al., 2016; Jakobi et al., 2018), four bare detectors were used to count thermal neutrons. In the second experiment on Oct 20, all plastic moderators had been temporarily removed from the epithermal units, to run thermal neutron detection with nine bare tubes and to observe potential self-shielding effects compared to otherwise shielded neighbouring detectors.

- **SN8:** The Styx Rover system from Styx Neutronica mounted in Van B (IVECO Daily) consists of eight stacked horizontal tube arrays, each holding 4–5 tubes of boron-lined detectors (Weimar et al., 2020). The system allows to read each module separately in order to observe potential asymmetrical effects. In two experiments on Oct 20, removable gadolinium shields were used to assess the neutron response along a track with and without a thermal neutron barrier.

Table 2 provides an overview of the various roving setups applied on October 19 and 20. The mobile detectors accumulated neutron counts over a record interval of 10 seconds. For a sufficiently high spatial resolution, the speed was set between 10 and 100 meters per minute. Inside the Wüstebach catchment, the areas accessible for car-borne roving were limited to the forest track crossing the catchment from southeast to northwest, along its northwestern border and to the main road (B258) in the south. Once, the HI rover was carried by hand across the catchment valley from west to east in order to capture a possible soil moisture gradient in that direction.





**Table 2.** Rover campaigns on the two intensive measurements campaigns, using different vehicles and detector modules (HI1: Hydroinnova unit (UFZ), HI9S: Hydroinnova System, $9\times$ units (FZJ), SN8: Styx Neutronica, $8\times$ units (UFZ)) which support various modes (E: epithermal, T: thermal, $_v$: vertical, $_h$: horizontal, E\T: Epithermal with extra thermal gadolinium shield. $^2$: all moderator shields removed from epithermal units.

| Day | Time | Vehicle | Rover | Epithermal | Thermal | Gd-Shield |
|---|---|---|---|---|---|---|
| Oct 19 | 06:45–08:01 | Blimp | HI1 | $E_h$ | | |
| | 09:10–12:07 | Car | HI1 | $E_h$ | | |
| | 12:30–13:50 | Van A | HI9S | $E_{hv}$ | $T_{hv}$ | $E_v$\T |
| | 15:02–16:36 | Blimp | HI1 | $E_h$ | | |
| Oct 20 | 07:50–10:10 | Van A | HI9S | $E_{hv}$ | $T_{hv}$ | $E_v$\T |
| | 08:50–10:15 | Car | HI1 | $E_h$ | | |
| | 10:55–12:45 | Van A | HI9S | | $T_{hv}^2$ | |
| | 11:15–14:15 | by hand | HI1 | $E_h$ | | |
| | 14:34–15:10 | Van B | HI1, SN8\T | $E_h$ | | $E_h$\T |
| | 15:42–16:42 | Van B | HI1, SN8 | $E_h$ | | |

## 3.6 Airborne Roving CRNS

Since ground-based CRNS roving is limited by the accessibility of areas and affected by local bias due to the road effect, we pioneered airborne neutron detection with a hot-air blimp. The blimp service was rented from the Airgraphics company, and the detector unit HI was mounted on a wagon at the bottom of the blimp in which the pilot is located. The airship could only be operated safely under calm wind and free sight conditions, i.e. in the early morning and in the late afternoon of Oct 19. The initial route design suggested to traverse the whole catchment area in dense sinuous lines at low and constant altitude. However, unpredictable wind conditions forced us to regularly adjust direction and the altitude, and to limit the measurement time to about one hour per flight. Therefore, the measurements focused on few key areas and gradients across the catchment. Additionally, a controlled vertical ascent was conducted to review the height dependency of the neutrons at a single location.

## 3.7 Hydrogravimetry

The basic concept of terrestrial gravimetry is to measure variations of gravity in time and space as a function of the mass distribution and its variations above and below the terrain surface. As modern gravimeter types are sensitive enough to capture water mass changes in their surroundings, several hydrological applications of this technique have been reported (see e.g., Van Camp et al., 2017, for an overview). A gravimeter senses all mass changes in its near-field surroundings of several hundreds of meters up to few kilometers in an integrative way, including storage variations in the unsaturated zone, the groundwater and, if applicable, in the snow pack.



For the JFC-2020, a gPhoneX gravimeter (Serialnumber: 151, Manufacturer: Micro-g LaCoste, USA) was deployed. The gPhoneX is a relative gravimeter based on the concept of springs (Niebauer, 2015) with a precision of 1 μGal ($10 \, \text{nm/s}^2$), an instrumental drift of $< 500 \, \text{μGal}$ per month, and a recording interval of one second. The instrument was deployed in a dedicated field enclosure for outdoor operation, the gPhone SolarCube (Reich et al., 2019, see Fig. 1). This energy self-sufficient mobile

container has an integrated pillar, based on a small concrete foundation at a depth of about 80 cm below the terrain surface. On top of the pillar, the automatic leveling platform "ODIN" was installed on which, in turn, the gravimeter was placed. The small footprint of the container (2 by 2 meters) and the deployment of the gravimeter sensor at a height of about 124 cm above the terrain surface increase the sensitivity of the instrument to near-surface soil moisture variations, similar to Güntner et al. (2017).

The gravity data were processed following state-of-the-art community standards (Crossley et al., 2013; Van Camp et al., 2017). This includes decimation to minutely and hourly time intervals, site-specific corrections, and the removal of all local and global signal components that are not of interest for the hydrological interpretation. Site-specific corrections included the removal of maintenance intervals, earthquakes and offsets, while large-scale corrections remove tidal (parameters by http://holt.oso.chalmers.se/loading/), global hydrological (ERA5, Hersbach et al. (2020)), global atmospheric (ATMACS,

Klügel and Wziontek (2009)) and non-tidal ocean loading effects (OMCT6, Dobslaw et al. (2017)). In absence of a sufficient number of absolute gravity measurements at the site for determining the instrumental drift of the the gPhoneX, the drift correction was carried out directly based on the gravimeter records as follows: After field experiments during the JFC-2020 period that included the short-term occupation of the gravimeter pillar with another instrument (not described within this publication), an exponential drift component on top of the long-term linear drift was observed in the gPhoneX time series. Thus, we split the

full time series into subsets between these experiments, and fitted a function that consists of an exponential and a linear term to each subset. This drift was then removed from the original data. Additionally, the reduction of the tidal signal with the synthetic tide parameter model resulted in a remaining tide-related signal component in the time series. After a Fourier analysis, this component with dominant frequencies between $1.1 \cdot 10^{-5}$ and $1.2 \cdot 10^{-5}$ Hz was removed by applying a notch-filter. Along with the raw gPhoneX records, the resulting hourly time series of gravity residuals is provided as part of this publication. It

represents the gravity effect of local hydrological storage dynamics, as well as residual errors related to the reduction of the other environmental and instrumental effects.

### 3.8 Local observation of soil water content and other soil data

The quantification of soil moisture from the measured neutron counts usually requires site-specific calibration, which in turn requires independent observations of soil moisture across the sensor footprint. For the presented dataset, the backbone for these

reference measurements is a permanent wireless soil sensor network (SoilNet) which is part of the TERENO observational infrastructure (Bogena et al., 2018, section 3.8.1). However, the SoilNet does not cover the catchment part south of the main road (B258). For those parts of CRNS footprints that were not sufficiently covered by SoilNet nodes, we conducted manual measurements with soil cores and FDR probes on October 19, 2020 (section 3.8.2). Fig. 2 provides an overview of SoilNet nodes and manual sampling locations. As our ambition was to provide a vertical soil moisture profile at the vicinity of each





CRNS sensor, the CRNS sensors were placed close to existing SoilNet nodes. As this was not possible for CRNS sensor 6 (south of the main road outside the SoilNet area), the sensor location was supplemented with one profile probe (see 3.8.3).

### 3.8.1 Wireless soil sensor network

The wireless soil sensor network (SoilNet) in the Wüstebach area has been operational since 2009. It consists of 600 ECH$_2$O EC-5 and 300 ECH$_2$O 5TE sensors (Decagon Devices; Rosenbaum et al., 2010) which measure soil moisture and temperature
every 15 minutes at nominally 150 locations, with two sensors offset by 10 cm at measuring depths of 5, 20 and 50 cm. The determination of apparent permittivity from the recorded raw data as well as the conversion from permittivity to volumetric soil moisture were documented in detail by Bogena et al. (2010). In this data publication, we provide a quality-controlled product (Wiekenkamp et al., 2016b, 2020) which provides soil moisture at an hourly resolution, and as an average of the two sensors at each depth. In addition to the quality control, remaining spurious data were flagged based on various criteria, including
plausible value ranges and plausibility of temporal dynamics. The time series are provided as one tab-separated (csv) file per measurement depth. It should be emphasized that, due to a general overhaul, the number of active SoilNet nodes during the JFC-2020 campaign amounted to 101 sensors (in contrast to the 150 nodes that had been originally installed).

The original 15 minute data including both sensors per depth are available in the TERENO data portal (section 4). Further technical and scientific documentation of the SoilNet observations is provided by a large body of literature, including e.g.
Bogena et al. (2010, 2018), Rosenbaum et al. (2010, 2012), and Wiekenkamp et al. (2016b, a), to name only a few.

### 3.8.2 Manual measurement of soil moisture and soil sampling

The manual measurements of soil moisture were conducted on October 19, 2020, in order to provide reference observations of soil moisture for the CRNS footprint areas without SoilNet coverage. In addition, sampling was conducted along a transect from east to west through the catchment (Fig. 2), carrying out manual measurements in the direct vicinity of SoilNet nodes.
All other measurement locations were surveyed with dGPS.

At 18 locations, measurements were carried out by extracting soil cores with cylinders at depth increments of 5 cm, starting at the surface, down to a measurement depth of 30 cm. The gravimetric water content of the samples was obtained in the lab, which involved oven drying at 105 °C and subsequent weighing. Residual water content and organic matter were determined by exposing composite samples to temperatures of 400 and 1000 °C for a duration of 16 and 12 hours. The composite samples
were produced for each measurement depth, and for four classes which resulted from the combination of the two dominant soil types Planosol and Cambisol and the two land use classes forest and shrubland/grassland.

At 50 locations, soil moisture profiles were obtained by using handheld ML2 ThetaProbes (Delta-T Devices LLC, Cambridge, UK). Measurements were carried out in vertical bore holes, incrementally drilled from the surface to a depth of 30 cm in steps of 5 cm (the depth corresponds to the electrodes' upper end after full insertion of the probe). In order to account for
small-scale variability, measurements were taken three times at each depth, with a slight rotation after each time. The recorded sensor voltage was, after calibration in air and water, converted to permittivity (see Fersch et al., 2020, for details), and from permittivity to volumetric soil moisture $\theta$. For the latter, we used the equation proposed by Zhao et al. (2016), but with adjusted



coefficients obtained by fitting the equation to thermo-gravimetric reference measurements of $\theta$ (see accompanying meta data for details).

### 3.8.3   Soil moisture profile probe

For the JFC-2020, we installed one soil moisture profile probe directly at the location of sensor 6 (see Fig. 2). We employed an FDR-based profile probe PR2/4 SDI (Delta-T Devices LLC, Cambridge, England, UK), which measures at 10, 20, 30, and 40 cm depths with a custom calibration. See Fersch et al. (2020) with regard to the conversion from the raw voltage readings (every 20 minutes) to volumetric soil moisture.

### 3.9   Lysimeter observations

Six high precision weighable lysimeters are operated at the location of CRNS 12 and the gravimeter. They are placed in a hexagonal design around a central service unit, including a weather station (Bogena et al., 2018), and are part of the TERENO SoilCan network (Pütz et al., 2016). The weight measurements are recorded every minute, other sensor parameters at 10-minute intervals.

Each stainless steel lysimeter (with a surface area of $1.0\,\mathrm{m}^2$, and a length of $1.5\,\mathrm{m}$), is filled with a monolithic soil core, which was taken from the same location within the Wüstebach catchment. After the filling procedure, a suction rake was inserted into the lysimeter bottom (1.45 m depth) consisting of six porous tubes. The weighing precision of each lysimeter is 10 g. The cumulative leachate volume is collected in a separate tank placed on a balance. The vegetation of the lysimeters corresponded to a representative section of the test site and can be classified as a forest meadow, whereby no plant cultivation was carried out. The main species are *Agrostics capillaris* and *Galium saxatile*, with a thick layer of moss (*Rhytidiadelphus squarosus*) covering the soil surface of the lysimeters (Groh et al., 2019). Since lysimeter weight dynamics, as a representative of soil water dynamics, are also affected by external disturbances (such as maintenance, wind, animals, etc.), the time series were subjected to various post-processing steps, including i) visual and automatic plausibility checks and ii) application of the adaptive window and adaptive threshold filter (AWAT,  Peters et al., 2017). Further details on instruments, lysimeter design, and suction control can be found in Pütz et al. (2016), Bogena et al. (2018), and Groh et al. (2018).

### 3.10   Vegetation and biomass

Epithermal and thermal neutron count rates are affected by all hydrogen pools within the footprint. In order to investigate the variability of neutron count rates in space, e.g. in the context of CRNS roving or a dense CRNS network, the spatial distribution of hydrogen in the vegetation biomass has to be characterized.

The major land cover type in the area is spruce forest (*Picea abies*). The age structure of the forest a rather homogeneous (planted around 1946 after comprehensive clearances), hence the spatial heterogeneity of the forest biomass is low in comparison to more structured and diverse forests. The central clear-cut area is characterized by various shrub species which are typical for the current stage of succession. Based on the vegetation mapping as well as the analysis of aerial photographs, four



land cover classes were defined: roads (no biomass), forest, grassland (dominantly herbaceous plants and small shrubs) as well
as shrubland (governed by a more advanced stage of succession with larger shrubs and some small trees).

For the quantification of above-ground dry biomass (Schmidt, 2021), we randomly selected 15 plots in the forest area and
15 in areas classified as grassland *or* shrubland area. On these 30 plots (12.5 m radius), the following procedures were carried
out to quantify the biomass in trees, shrubs, herbaceous plants and grasses, as well as in deadwood:

– Trees: For all trees in a plot, we measured the breast-height diameter and the height (for the latter, a TruePulse laser
range finder, manufactured by Laser Technology, was used). The tree biomass was computed from these variables by
         using species-specific allometric functions (Zell, 2008). For trees with a height smaller than 5 m, allometric functions
         from Riedel and Kändler (2017) were applied.

– Shrubs: The main species sampled as shrubs were rasperry (*Rubus idaeus*), blackberry (*Rubus sect. Rubus*), blueberry
         (*Vaccinium myrtillus*) and Scotch broom (*Cytisus scoparius*). Dry shrub mass per plot was estimated by using allometric
functions from Bolte (2006) which employ the species-specific average shrub height and coverage per plot.

– Herbaceous plants: in each of the 30 plots, we sampled herbaceous plants, but also above-ground litter, within three
         30x30 cm squares, and determined the dry biomass after drying to constant weight at a temperature of 70 °C.

– Deadwood: length, diameter and level of decomposition were recorded for all deadwood pieces in the plot, and dry
         biomass was obtained based on dry mass density values for spruce deadwood published by Kahl (2003).

That way, dry above-ground biomass inventories were obtained for each plot and biomass pool (trees, shrubs, herbaceous,
and deadwood). We then averaged and aggregated these pools for each of the land cover classes, and came up with the following
estimates: $0\,\mathrm{kg\,m^2}$ for the roads, $27.9\,\mathrm{kg\,m^2}$ for forest, $4.6\,\mathrm{kg\,m^2}$ for grassland, and $7.7\,\mathrm{kg\,m^2}$ for the shrubland area. The
respective areal extent was mapped in the field and is provided in the data set.

## 4   Relevant data provided by third parties

This section highlights relevant data sets which are not included in this publication, but were published before or provided by
other organisations or channels.

### 4.1   Previously published soil data

Gottselig et al. (2017a) already published an extensive dataset on soil properties at 155 sampling locations in the Wüstebach
area, including soil organic carbon content, bulk density, soil texture and soil chemical parameters across different depths. The
full dataset is available at the TERENO data portal (Gottselig et al., 2017b).



## 4.2 Incoming neutron flux

Variations of the incoming cosmic-ray neutron flux were recorded by neutron monitors. The corresponding data can be downloaded from the Neutron Monitor Database, http://www.nmdb.eu. Based on suggestions from previous studies (Hawdon et al., 2014; Schrön et al., 2016; Baatz et al., 2015; Jakobi et al., 2018), the neutron monitor at Jungfraujoch (JUNG) can serve as
reference for the incoming neutron flux in the Wüstebach catchment.

## 4.3 Hydro-meteorological observations in the TERENO data portal

Various hydrometeorological and hydrological data for the TERENO Eifel/Lower Rhine Valley Observatory, specifically for the Wüstebach catchment, are available for the study period of the JFC-2020 at https://www.tereno.net/ddp/ (TERENO Data Portal), including comprehensive metadata. Of specific interest for users of the published data set, we consider the following
observations (see also Fig. 2b):

- Observations of various meteorological variables, including precipitation, air temperature, relative humidity, and barometric pressure are recorded at three climate gauges in the study area at a 10 minutes interval: gauge WU_KR_002 and WU_KR_001 in the clear-cut area, and gauge WU_BKY_010 at the lysimeter station. Furthermore, two eddy-flux towers are available, one located in the northern forested part (WU_EC1), and another in the central clear-cut area (WU_EC2).

- Two runoff gauges record discharge at a 10 minutes interval: WU_AW_14 at the catchment outlet, and WU_AW_10, about 100 m upstream.

- Four continuous groundwater level gauges are available in the inner valley (TERENO IDs WU_GW_001, -3, -5, and -9);

- SoilNet data for soil water content and soil temperature at 150 nodes are available at a 15 minutes interval and at depths of 5, 20 and 50 cm (see section 3.8.1 for further details).

## 4.4 Terrain and soil maps

A highly resolved digital elevation model (DEM) can be helpful in various application contexts with the presented dataset, e.g., for the hydrological analysis of runoff concentration or the estimation of the ground water depth. A DEM at 1 m resolution and a vertical accuracy of 20 cm can be obtained from the open data portal of Nordrhein-Westfalen (https://open.nrw). At the same open data portal, a soil map at a scale of 1:50,000 (BK50) is available for the entire federal state of NRW. A more
detailed soil map at a scale of 1:2500, using the World Reference Base for soil classification, was prepared by Richter (2007). The map is not available online, but has to be purchased from the Geological Survey of North Rhine-Westphalia, Germany (https://www.gd.nrw.de/).



## 4.5 Land use, roads, waterways

During fieldwork and for visualisation, we used and complemented OpenStreetMap data layers (OpenStreetMap contributors, 2021) available via http://download.geofabrik.de, namely landuse, waterways, and traffic ways. The data are distributed under ODbL license (www.openstreetmap.org/copyright).

## 5 Cross-scale soil moisture patterns in space and time: an exemplary view at the data

This paper is not about the scientific analysis of the presented dataset. Still, we provide an exemplary view at the data in order to convey an idea of spatial and temporal soil moisture patterns, as well as of differences between sensors at different horizontal and vertical scales. To that end, we used standard procedures to convert neutron intensities (observed by the stationary dense CRNS cluster and CRNS roving) to soil moisture, and to put these into context with observations from the SoilNet, the gravimeter, six weighable lysimeters, rainfall as well as discharge. Furthermore, we provide, for the very first time, exemplary visualisations of the airborne CRNS roving observations. Please note that we only provide a short outline of required data processing steps. This is because we just give examples of how to put the different observations into context, while a comprehensive analysis is subject to future research.

**SoilNet observations as soil moisture reference**

The SoilNet represents the main ground truth reference in this data set. The observations were averaged at a daily resolution and then interpolated to a 10x10 m grid at each measurement depth (5, 20, 50 cm) using Ordinary Kriging (exponential variogram model with a range parameter of 150 m, see Bogena et al., 2010). Please note we excluded the catchment part south of the main road (B258) which is not covered by the SoilNet. To obtain a soil moisture value that is comparable to the soil moisture obtained from the CRNS measurements, we computed, for each grid cell, a vertically-weighted average (see Schrön et al., 2017, for a specification of weighting functions). The corresponding patterns are shown at three selected dates in the upper panel of Fig. 3: On September 24, the soils were comparatively dry, as there had been almost no rain for several weeks (see dashed vertical lines in Fig. 4b). Within the following three days of rainfall, the soil moisture rose steeply until September 27. On October 28, the catchment wetness is even higher as a result of further rainfall and autumnal cooling.

The SoilNet clearly displays a higher soil moisture in the clear-cut area (as already found by Wiekenkamp et al., 2016a), possibly due to the influence of both groundwater and sparse vegetation cover. Already for the dry date (September 22), marked wet spots were present in the central and northern parts, in close proximity to the stream. Wetter areas also extended to the south-west between CRNS sensors 12 and 10 where the vegetation is less dense. Towards the end of October, the wet areas started to cover the entire clear-cut area, but the soil moisture on the forested hillslopes and hilltops increased, too.

Fig. 4b highlights the resulting temporal dynamics of the catchment-scale soil moisture as obtained from the SoilNet observations. At September 24, the average catchment wetness reaches its minimum over the campaign period after three weeks of



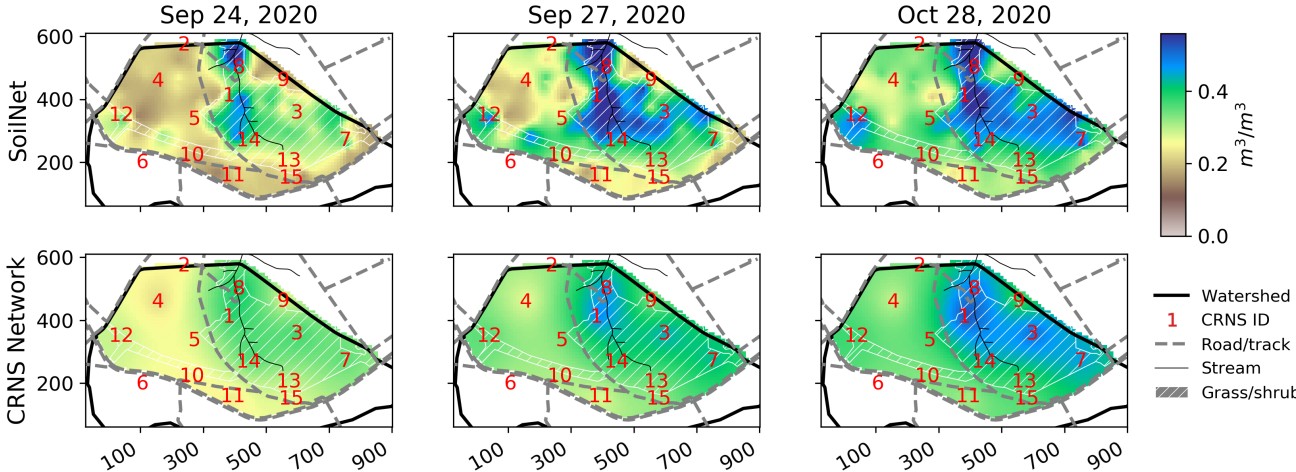

**Figure 3.** Maps of soil water content as obtained from the interpolation of SoilNet (upper panel) and stationary CRNS observations (bottom panel), for three different dates (dry, intermediate, wet). The SoilNet data represent a weighted average over three measurement depths that intends to mimic the penetration depth of the CRNS data. OSM layers were used to represent roads, landuse, and waterways (© OpenStreetMap contributors, 2021. Distributed under the Open Data Commons Open Database License (ODbL) v1.0.).

almost no rain. Thenceforth, the catchment wetness increases more or less continuously until the end of the campaign, with a particularly steep increase between September 24 and 27.

**Soil moisture estimation from stationary CRNS**

We used the observations of the CRNS network to represent the distribution of soil water content in space and time. To that end, we followed a similar procedure as outlined by Heistermann et al. (2021b): **(i)** we used the detector sensitivity (see Tab. 1) to standardize neutron intensities of all sensors to a common level (of the "calibrator" probe); **(ii)** as outlined by Andreasen et al. (2017), neutron intensities were corrected for the flux of incoming cosmic neutrons (section 4.2) as well as of barometric pressure and atmospheric water vapor (section 4.3); **(iii)** to convert neutron count rates to soil moisture, we estimated, for each CRNS sensor $i$, a calibration parameter $N_{0,i}$ (Andreasen et al., 2017) (using SoilNet observations available in each footprint on October 19, 2020, which were vertically and horizontally weighted according to Schrön et al. (2017)); please note that the effects of biomass were not yet explicitly accounted for, however, the calibration of $N_{0,i}$ implicitly takes biomass into account; **(iv)** for the entire study period and each CRNS sensor $i$, volumetric soil moisture estimates were obtained from daily average values of the observed neutron count rates by using the estimated calibration parameters $N_{0,i}$; **(v)** the daily soil moisture estimates were interpolated to a 10x10 m grid using Ordinary Kriging (same variogram model as for the SoilNet interpolation).

The resulting spatial patterns are shown in the bottom panel of Fig 3. As expected, the spatial distribution of CRNS-based soil moisture is much smoother, mainly as a result of the large sensor footprint. However, at the hectometer-scale, the soil moisture patterns generally compare well to the patterns acquired with the SoilNet. The results shown in Fig. 4b are consistent

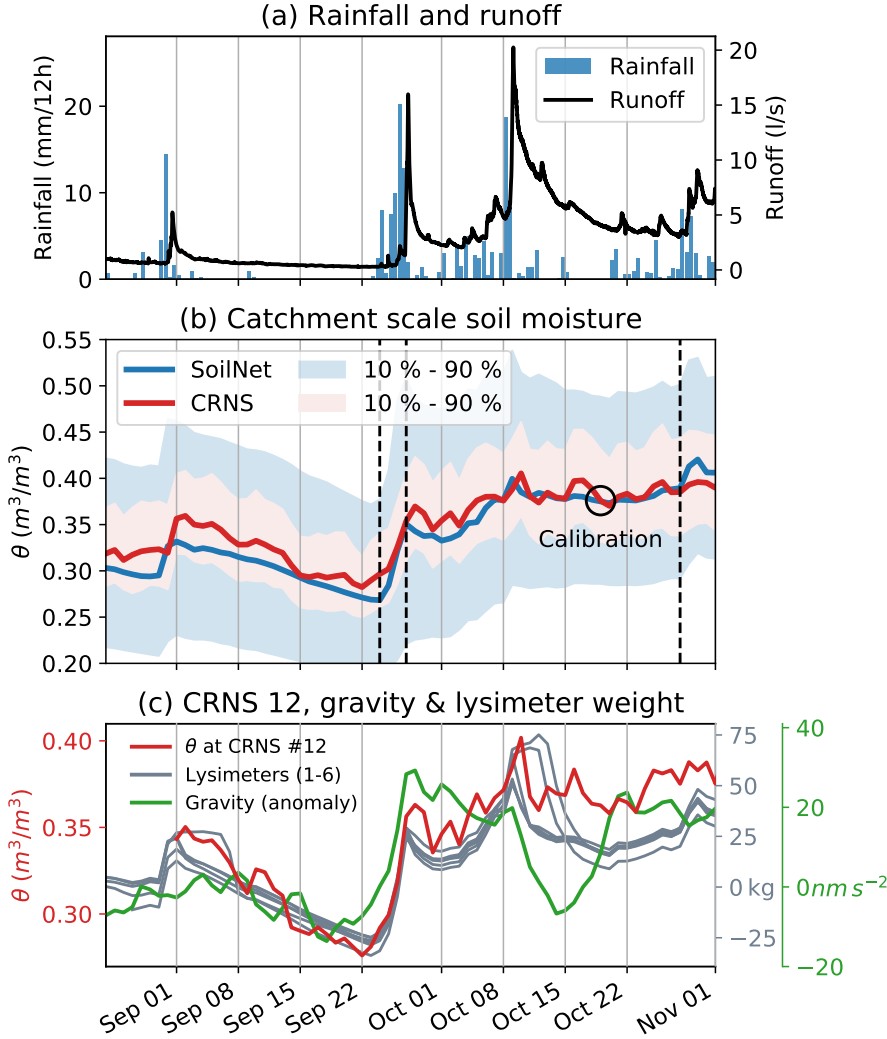

**Figure 4.** Temporal dynamics of selected parameters during the JFC-2020 campaign. (a) Rainfall (in mm/12 hour intervals) at TERENO climate station (ID WU_KR_002), and runoff (in l/s) at the catchment outlet (TERENO runoff gauge ID WU_AW_014); (b) Catchment scale soil moisture as obtained from the SoilNet and the CRNS network after spatial interpolation (daily average values, see main text for further explanation); solid lines represent the mean areal soil moisture, the colored shadows the inner 80 percentiles of the gridded values; the black dashed lines mark the dates shown in Fig. 3; the circle indicates the date of CRNS calibration; (c) dynamics of weight anomalies for the six lysimeters (weight difference to the average weight of a lysimeter), together with the observed local residual gravity anomalies, and the soil moisture retrieved for CRNS 12 (see Fig. 2 for locations). All displayed values in subplot C are daily averages.

with this notion: While the range between the 10th and 90th percentiles of the soil moisture distribution as obtained from the SoilNet is considerably larger than for the CRNS-based soil water content, the temporal development of the catchment-wide soil moisture means from SoilNet and CRNS agree quite well. In order to utilize the CRNS network for improving



the representation of heterogeneity within the catchment, prospective research should test and further develop the inversion technique suggested by Heistermann et al. (2021b), and should use the acquired biomass data to account for hydrogen in the vegetation, and hence to allow for the estimation of a uniform $N_0$ value across all sensors (instead of one value for each sensor).

**Moisture-induced gravity variations**

Fig. 4c shows the specific situation around CRNS sensor 12 which was installed right beside the gravimeter and the lysimeter station. The residual gravity anomalies are similar to the general dynamics of the lysimeter weights and the soil moisture of CRNS 12, consistently representing the storage increase from the strong rainfall event in late September. After that, both CRNS and lysimeters indicate a further increase in soil moisture until about October 9, while gravity already decreases during this period, followed by a pronounced decrease between October 9 and 14 - which, in turn, is only partly visible in the CRNS and lysimeter data. It will be subject to prospective research whether the differences between these three monitoring methods indicate flow processes from the near-surface to the deeper unsaturated zone, or in the groundwater, or whether they are caused by external non-hydrological factors that might have been inadequately reduced from the residual gravity time series. To this end, future work should also include the explicit forward modelling of the gravity effect of water storage variations at different depths, as well as a comprehensive water balance at the catchment scale.

**Spatial soil moisture patterns from CRNS roving**

Intensive CRNS roving was carried out on October 19-20, 2020, in the Wüstebach catchment (section 3.5), in order to add spatial detail along selected transects.

Fig. 5 shows four examples of the various measurements that took place (see Table 2). For this figure, the observed neutron intensities were averaged along the roving track over 2 minutes (rolling mean) and a 20 m radius ($W_r$-distance weighting based on Schrön et al. (2017)) to improve the signal-to-noise ratio. The data also underwent basic correction for the effects of atmosphere, soil properties, as well as road conditions, as outlined in Schrön et al. (2018). The underlying road network properties of the study area are part of the published roving data set.

The thermal neutron signal (Fig. 5a) measured by the HI9S rover indicates a strong correspondence to the vegetation cover along the track: lower counts in the southern and central part of the transect adjacent to the grassland parts with lower biomass pools as compared to the sections through the forested parts. This is consistent with Jakobi et al. (2018) and could be later employed for vegetation correction, using the observed biomass distribution (section 3.10) as a reference. The soil moisture, as obtained from epithermal neutron intensities observed by the different systems and designs, is displayed in Fig. 5b (HI9S rover), Fig. 5c (HI1 rover), and Fig. 5d (SN8 rover). Without comprehensive corrections, all these soil moisture patterns will not only be affected by the actual soil moisture distribution, but by the effects of vegetation and soil, too. The differences, however, between these three patterns are assumed to be caused by the individual detector designs and shielding options (see Tab. 2). Hence, these data will be useful in future studies to better understand the influence of such design-related aspects.



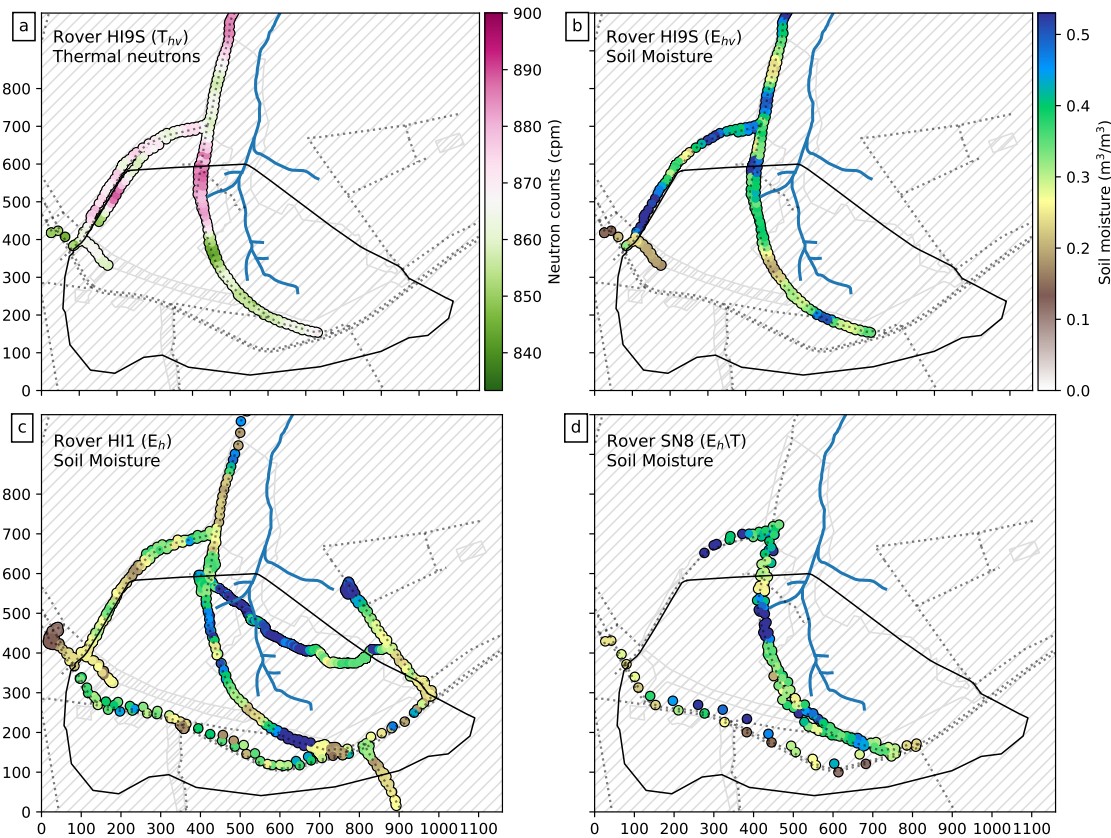

**Figure 5.** Exemplary data from mobile CRNS rover measurements on October 19 and 20. (a) Thermal neutrons measured by the HI9S rover, (b) derived soil moisture from HI9S rover, (c) derived soil moisture from the HI1 rover in a car, including the track through the central grassland valley (hashed area), carried by two persons, (d) derived soil moisture from the SN8 rover (with gadolinium shielding to prevent thermal neutrons from entering the detector signal).

**Airborne roving**

Neutron detection on a blimp is a promising technique to capture patterns of albedo neutron intensity in inaccessible terrain. The landing site is slightly outside of the shown spatial extent in the North of the catchment. While constant height above ground and full coverage of the catchment was initially intended, the steering of the blimp was affected by unpredictable changes of wind speed and direction, which lead to deviations from the originally intended route. As can be seen in Fig. 6, the flight height has, in addition to vegetation and soil moisture, a substantial effect on the epithermal neutron count rate (Fig. 6).

In future studies, these effects have to be quantified and corrected for in order to enable a comparison to soil moisture patterns observed by other sensors in the campaign. To support such research, the landing site in the North was also used for a lifting experiment up to 170 m (see minutes 5–16 in Fig. 6b).



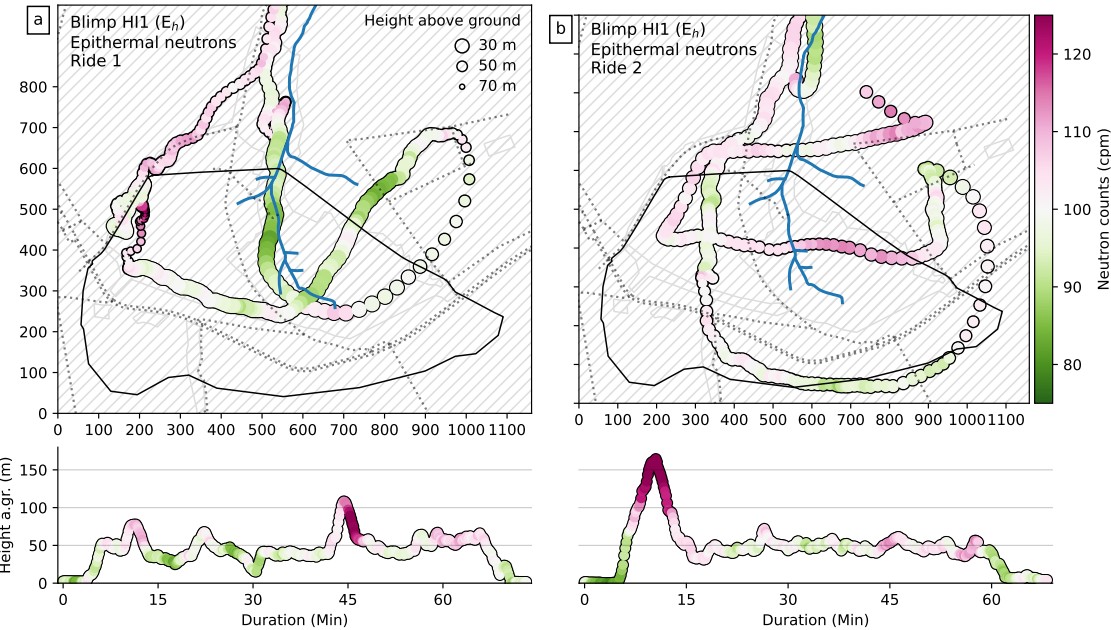

**Figure 6.** Neutron count rates observed by airborne CRNS from a blimp during two rides, (a) in the morning and (b) in the late afternoon (compare Table 2). The colored dots show neutron counts every 10 seconds along the trajectory. Neutrons were corrected for air pressure, humidity, and incoming flux, but still indicate strong influence of the flying height above ground and vegetation. The time series of this height is shown in the bottom panels. The height is also indicated by the size of the dots in the maps.

# 6  Conclusions

In the autumn of 2020, members of the DFG research group Cosmic Sense bundled their resources in instruments, expertise
and workforce to conduct a Joint Field Campaign (JFC) in which 15 CRNS sensors were deployed for more than two months in the 39 ha headwater catchment of the Wüstebach in the Eifel mountains (western Germany) with a focus on retrieving root zone soil moisture.

The resulting dataset constitutes a yet unique contribution that should be useful to various research communities.

Why is this data set *unique*? In comparison to the earlier JFC-2019, the JFC-2020 covered a different landscape and season:
the dominantly forested Wüstebach catchment is a more challenging environment for CRNS-based soil moisture retrieval, while the watershed is topographically more pronounced, facilitating hydrological model applications; the campaign period covered the autumnal re-wetting instead of the drying in summer. Furthermore, the CRNS sensor density was even higher (average density of 38 sensors per km$^2$ instead of 24 per km$^2$), with almost no gaps in time. Finally, the instrumentation featured a unique combination with an extensive wireless soil sensor network (for calibration and validation), a gravimeter,
weighable lysimeters, as well as ground-based and, for the first time, airborne CRNS roving.





We are confident that this data set will be a *useful* resource for various scientific purposes: to advance the estimation of soil water content from CRNS, e.g. by considering the role of vegetation; to study water storage variations beyond the root-zone by hydrogravimetry; to identify scale-specific combinations of sensors and methods in studying soil moisture variability across space and time (specifically by using overlapping CRNS together with inversion-like retrieval methods and auxiliary

information from e.g CRNS roving and hydrological modelling, see Heistermann et al., 2021b); to improve the understanding and simulation of land-atmosphere exchange as well as hydrological and hydrogeological processes at the hillslope and the small catchment scale; and to support root-zone soil moisture retrieval from airborne and spaceborne remote sensing platforms by using CRNS-based soil moisture estimates at various temporal and spatial resolutions.

In summary, this data set should support those who are interested in advancing CRNS and hydrogravimetry at the level of

single instruments as much as those who are interested in the broader picture of observation and simulation of soil moisture and hydrological processes from the point to the small catchment scale.

## 7    Code and data availability

For this data publication, we used EUDAT (https://eudat.eu). Within this data infrastructure, the services B2SHARE and B2HANDLE allow to share data, to manage identifiers, and to provide long-term persistence. Please see Heistermann et al.

(2021a) as a reference to the data repository at the B2SHARE service (https://doi.org/10.23728/b2share.afb20a34a6ac429ca6b759238d842765). Tab. 3 links the data subsets in the repository to the corresponding subsections of this paper. Each subset of data is accompanied by a text file with meta data (JSON format) which will also provide further information on the data format in case this is different from the specifications in Sect. 3.2.

**Table 3.** Structure of the data repository, and the relation of data subsets to the subsections of this paper.

| Section | Observation | Data subset in the repository |
|---------|-------------|-------------------------------|
| 3.3 | Stationary CRNS | crns_stationary.zip |
| 3.5 | Roving CRNS | crns_roving.zip |
|  | Roving onboard photos | crns_roving_camera.zip |
| 3.6 | Airborne CRNS | crns_airborne.zip |
|  | Airborne onboard photos | crns_airborne_camera.zip |
| 3.7 | Gravimeter (processed) | gravimeter_processed.zip |
|  | Gravimeter (raw) | gravimeter_raw.zip |
| 3.8.1 | SoilNet (WSN[*]) | soilmoisture_soilnet.zip |
| 3.8.3 | Profile probe | soilmoisture_profileprobe.zip |
| 3.8.2 | Manual soil sampling | soilmoisture_manual_sampling.zip |
| 3.9 | Lysimeter | lysimeter.zip |
| 3.10 | Vegetation / biomass | biomass.zip |

[*] Wireless soil moisture sensor network





*Author contributions.* The lead authors MH, HB, TF, AG, JJ, DR, and MS designed and conducted the JFC, processed the data and drafted the manuscript. MR processed the gravimeter data. AP, BF, CZ, DR, JJ, MS, SO, and TF participated in the JFC-2020 field work. TP and JG contributed and processed the lysimeter data for the campaign period. CZ and SZ supported the roving campaign and analysis. VD co-developed Fig. 1. SO suggested the idea to operate CRNS in a dense cluster and was the principal investigator in charge of the JFC-2020. All authors contributed to writing the manuscript.

*Competing interests.* No competing interests.

*Acknowledgements.* This research was funded by the Deutsche Forschungsgemeinschaft (DFG, German Research Foundation) – research unit FOR 2694 "Cosmic Sense", project number 357874777. Funding for the TERENO-Rur Hydrological Observatory infrastructure is provided by the BMBF (Bundesministerium für Bildung und Forschung) and the Helmholtz Association. We thank Christian Budach, Peter Biro and Sophia Dobkowitz (University of Potsdam) and Daniel Dolfus (Research Center Jülich) for their contributions to field work and sensor maintenance, and Sandra Szulz-Seyfried for supporting the coordination of the campaign. We would also like to thank the EUDAT services for providing an excellent data management infrastructure.



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
