# Peer review of "Soil moisture observation in a forested headwater catchment: combining a dense cosmic-ray neutron sensor network with roving and hydrogravimetry at the TERENO site Wüstebach"

_Earth System Science Data, 2021_

## Author Comment (AC1)

**Interactive Discussion: Author Response to Referee #1**

**Soil moisture observation in a forested headwater catchment: combining a dense cosmic-ray neutron sensor network with roving and hydrogravimetry at the TERENO site Wüstebach**

Maik Heistermann et al.
*Earth Syst. Sci. Data Discuss.,* `doi:10.5194/essd-2021-445`
* * *
**RC:** *Reviewer Comment*,    AR: *Author Response*,    ☐ Manuscript text

Dear Referee,

we would like to thank you very much for your willingness to review this paper, and for your very swift - and positive - response to the manuscript.

Please find our responses to your comments below. These should be considered as preliminary (part of the interactive discussion). The final implementation of changes also depends on another referee report that is still pending.

Thanks again for your efforts!

Kind regards,
Maik Heistermann
(on behalf of the author team)

**1.1.  CRNS Calibration**

**RC:** *[...] The only really major criticism I have concerns the calibration of the stationary cosmic ray neutron sensors (CRNS). It has been shown that highly variable systems like temperate forests require calibration for at least two points in time (one dry state and one wet state of the system, preferably), (see, e.g., Heidbüchel et al., 2016; Tan et al., 2020). Having the SoilNet sensors with continuous monitoring available, this can be done – however, the SoilNet does not cover the entire footprint of all the CRNS. Instead, one manual soil core sampling campaign was conducted to 'fill in the gaps' and get additional data on bulk density, lattice water content, etc. This allows for the proper calibration of the CRNS at (only) one point in time. I would expect a discussion of this deliberate choice to only use one point in time for calibration – in particular noticing the fact that in your case the calibrated CRNS seems to consistently overestimate soil moisture during the dry periods.*

AR:  We thank you for pointing out this important issue. In the original manuscript, we had decided not to address it in depth since the prime focus of the paper is on the presentation of the data set and some of its fundamental properties. Based on your comment, though, we agree that it makes sense to discuss the different calibration

options and the resulting implications for different use cases.

As you correctly pointed out, the calibration setup was a deliberate choice which was motivated particularly by enhancing the comparability to the SoilNet observations. You also already laid out the trade-offs we have to confront in this context:

- Using only the SoilNet for calibration reduces the number of data points per footprint, particularly for outward CRNS sensors. In turn, the resulting CRNS-based soil moisture estimates are more representative for the SoilNet area (i.e., north of the main road). That increases the comparability to the SoilNet observations, and hence the evaluation of the CRNS-based soil moisture estimates over space and time. That way, we could also introduce another calibration time for drier conditions, as you have pointed out.

- However, if we intend to estimate the mean soil moisture of the entire catchment (including the area south of the main road), we need to include the manual samples from the campaign on October 19, 2020. This would be advisable e.g. for hydrological applications that aim at closing the catchment's water balance. The downside is, as you pointed out, that these measurements are only available for one point in time, and that the comparability to the SoilNet pattern might be biased when additional points from outside the SoilNet area are included in the calibration.

One possible perspective to resolve this apparent conflict is to calibrate $N_0$ not individually for each sensor, but to estimate one single $N_0$ value for all sensor locations as demonstrated by [Heistermann et al., 2021]. This requires to carefully homogenize neutron intensities across sensors by accounting for the effects of sensor sensitivity, vegetation, soil carbon and lattice water, and would be possible for multiple calibration points in time, too. Such an approach would also allow to combine SoilNet observations and manual measurements. The validity of any single $N_0$ estimate could then be evaluated in a set of calibration/validation experiments, in order to demonstrate transferability in space and time (within the campaign period and within the study area).

There is ample room for experimenting with these options and their combinations in future studies. While this exceeds the scope of this data paper, we agree that the audience should be aware of these options and their implications. Hence, we will extend the paragraph on "Soil moisture estimation from stationary CRNS" (ll. 440-460) accordingly, and will also emphasize the benefit of calibrating at two (or more) points in time, as outlined above.

**1.2. Other specific comments**

**RC:** *Abstract, Line 7: Do you mean 'watershed boundary'? 'Watershed' alone can also mean the catchment.*

AR: In fact, we have used the term watershed, in this context, in the sense of the "watershed boundary". As this is obviously ambiguous, we will replace "watershed" by "catchment boundary".

**RC:** *Line 283-284, 307: In order to calibrate a CRNS properly it is recommended to independently measure soil moisture twice (preferably under wet and under dry conditions, see, e.g. Heidbüchel et al., 2016; Tan et al., 2020). Also, it is recommended to have 18 different sampling locations within the footprint of a CRNS (see Franz et al., 2012). For many of the CRNS this requirement is not fulfilled. How do you justify this? At least it would be added value to provide a two-point calibration.*

AR: Please see our response to your first comment 1.1.

**RC:** *Line 311: What are these 18 locations? In Figure 2 there are more than 18 locations marked with 'manual sampling'.*

**AR:** Thank you for pointing out this possible misunderstanding. As you stated correctly in the next comment, there were 68 manual measurements of soil moisture profiles: 18 measurements with soil cores (gravimetric determination in the lab, see ll. 311-316), and 50 in-situ measurements with handheld FDR-sensors (see ll. 317-324). We had decided not to distinguish the two approaches in Fig. 2 for the sake of clarity. However, as the you explicitly requested the positions of the soil core sampling (next comment), we will distinguish the two approaches in Fig. 2, and also clarify, in the first paragraph of Sect. 3.8.2, the total number of manual samples.

**RC:** *Line 317: I guess that is where the extra 'manual samples' derive from. I would mark them differently in Figure 2. (I really want to know where you did the soil core sampling).*

**AR:** Please see our response to the previous comment.

**RC:** *Line 357: Why won't you tell us the exact number of plots in the grassland and in the shrubland areas?*

**AR:** We will. Originally, we had intended to distinguish only two land cover types: forest and open land - which is why we assigned 15 plots to each type. However, during vegetation mapping and biomass sampling, it turned out that a few parts of the "open land" were already dominated by shrubs which is why we differentiated into grassland (with 11 sample plots) and shrubland (with four sample plots). We will add these numbers to the manuscript in the line which you referred to.

**RC:** *Line 409: What do you mean by 'complemented'?*

**AR:** By "complemented", we meant that we also contributed our improved land cover maps to the OSM community land use data. As this is not really relevant to the paper, we will just drop "and complemented" from the revised version of the manuscript.

**RC:** *Line 445-448: You only used one date (and one condition) to calibrate N0, although you have SoilNet values for all kinds of soil moisture conditions. That is dangerous and potentially weakens the measurement performance of the CRNS. You can see in Fig. 4b that the CRNS overestimates soil moisture in dry conditions – this could have been avoided with another calibration performed when it's dry.*

**AR:** Please see our response to your first comment.

**RC:** *Also, what about the manual samples from the soil cores, did you use them at all for calibration? If so, how did you incorporate them? If not, why not? After all, SoilNet does not cover the entire footprint of all of the sensors. At least, I would like to see this discussed.*

**AR:** In our exemplary data analysis in section 5, we did not use the manual sample for calibration, but focused on the SoilNet observations. We will explicitly discuss this choice as already elaborated in our response to comment 1.1.

**1.3. Technical corrections**

**RC:** *Abstract, Line 15: 'hillslope' instead of 'hill-slope'.*

**AR:** Will be corrected.

**RC:** *Abstract, Line 16: '...the retrieval OF soil water...'.*

AR:     Will be corrected.

**RC:**   *Line 39: 'Soon enough...'?*

AR:     We will replace "Soon enough [...]" by "Soon after the feasibility of soil moisture observation with stationary CRNS had been demonstrated, a mobile CRNS sensor [...]"

**RC:**   *Line 41: Schrön et al., which year?*

Thanks for spotting the broken reference. We will fix it:

Schrön, M. et al. (2021). Neutrons on rails: Transregional monitoring of soil moisture and snow water equivalent. Geophysical Research Letters, 48, e2021GL093924. doi:10.1029/2021GL093924

**RC:**   *Line 350: '...of the forest IS rather homogeneous...'*

AR:     Will be corrected.

**RC:**   *Line 363: raspBerry.*

AR:     Will be corrected.

**RC:**   *Line 402: '...groundwater depth...' (no space between ground and water).*

AR:     Will be corrected.

**RC:**   *Line 403: North Rhine-Westphalia is the English word.*

AR:     Will be corrected.

**RC:**   *Line 438: 'Thenceforth' is quite archaic and literary. I have never seen it used in a scientific paper.*

We will replace "thenceforth" by "thereafter".

**References**

[Heistermann et al., 2021] Heistermann, M., Francke, T., Schrön, M., and Oswald, S. E. (2021). Spatio-temporal soil moisture retrieval at the catchment scale using a dense network of cosmic-ray neutron sensors. *Hydrology and Earth System Sciences*, 25(9):4807–4824.

---

## Author Comment (AC2)

**Interactive Discussion: Author Response to Referee #2**

**Soil moisture observation in a forested headwater catchment: combining a dense cosmic-ray neutron sensor network with roving and hydrogravimetry at the TERENO site Wüstebach**

Maik Heistermann et al.
*Earth Syst. Sci. Data Discuss.,* `doi:10.5194/essd-2021-445`
* * *
**RC:** *Reviewer Comment*,    AR: *Author Response*,    ☐ Manuscript text

Dear Referee,

we would like to thank you very much for your willingness to review this paper, and for your positive and constructive response to the manuscript and the data set.

Please find our responses to your comments below.

Kind regards,
Maik Heistermann
(on behalf of the author team)

**Specific comments**

**RC:** *L19 – suggest "...limited, particularly when small scale variability is high."*

 AR:  Thanks for the suggestion. Maybe we can even shorten this sentence further so it becomes:

> "The spatial representativeness of conventional point-based soil moisture measurements is often limited by high small-scale variability [...]".

**RC:** *L23 – replace "confronted" with "limited by"*

 AR:  Will be implemented accordingly.

**RC:** *L32 – suggest "…100-150 m with a vertical depth…"*

 AR:  Will be implemented accordingly.

**RC:** *L39 – suggest replacing "soon enough" with "More recently"*

 AR:  As a response to a comment by referee 1, we suggest the following:

> "Soon after the feasibility of soil moisture observation with stationary CRNS had been demonstrated, a mobile CRNS sensor ("CRNS roving") was established [...]".

We prefer to keep "soon" instead of "more recently" because [Desilets et al., 2010] already published the first proof-of-concept for CRNS roving two years after [Zreda et al., 2008]. To clarify this, we will add the reference "[Desilets et al., 2010]" after this sentence.

**RC:** *L83 – fix "between 10 and 170 m"*

AR: Will be fixed.

**RC:** *L105 – suggest "It is available via EUDAT (see Heistermann et al., 2021a)."*

AR: Will be implemented accordingly.

**RC:** *L113 – suggest "was required to;" then remove "to" from the start of each point*

AR: Will be implemented accordingly.

**RC:** *Figure 2 – would fit better after dot point rather than within*

AR: We agree. However, figure positioning will be entirely revised during type setting / copy editing with a two column format, so we'd prefer not to interfer at this point.

**RC:** *Table 1 – would fit better after the dot points rather than within*

AR: Please see our response to the previous comment.

**RC:** *L350 – 2nd sentence makes no sense*

AR: We will fix the error, so the sentence(s) will read:

> "The age structure of the forest is rather homogeneous as it was planted around 1946 after comprehensive clearances. Hence the spatial heterogeneity of the forest biomass is low in comparison to more structured and diverse forests [...]"

**RC:** *L412 and L413 – I am not sure exemplary is a good word to use. Do you just mean an example of data usage?*

AR: We agree that "exemplary" is not ideal. It was meant in terms of "non-exhaustive" / "by-example". We were not aware of the meaning of "exemplary" in the sense of "best practice". We hence suggest to

- change the header of section 5 (l. 412) to "Examples of cross-scale soil moisture patterns in space and time"

- change l. 413 to "[...] we will provide selected examples to convey an idea of spatial and temporal soil moisture patterns as well as of differences between sensors at different horizontal and vertical scales."

**RC:** *Figure 6 caption – suggest "flights" rather than "rides"*

AR: Will be changed accordingly.

**RC:** *L524 change to "... B2HANDLE allow users to share . . . "*

AR: Will be changed accordingly.

**References**

[Desilets et al., 2010] Desilets, D., Zreda, M., and Ferré, T. P. A. (2010). Nature's neutron probe: Land surface hydrology at an elusive scale with cosmic rays. *Water Resources Research*, 46:W11505.

[Zreda et al., 2008] Zreda, M., Desilets, D., Ferré, T. P. A., and Scott, R. L. (2008). Measuring soil moisture content non-invasively at intermediate spatial scale using cosmic-ray neutrons. *Geophysical Research Letters*, 35(21):L21402, 1–5.

---

## Author Response (AR1)

**Author Response to Referee Comments**

**Soil moisture observation in a forested headwater catchment: combining a dense cosmic-ray neutron sensor network with roving and hydrogravimetry at the TERENO site Wüstebach**

Maik Heistermann et al.
*Earth Syst. Sci. Data Discuss.,* `doi:10.5194/essd-2021-445`
* * *
**RC:** *Reviewer Comment*,     AR: *Author Response*,     ☐ Manuscript text

Dear Editor, dear referees,

we would like to thank you for the positive and constructive feedback to our manuscript and the corresponding data set, and for the time and resources you have invested in this review process.

Herewith, we would like to submit a revised version of the manuscript. Please find our responses to both referees below. These are very close to our responses in the interactive discussion, but now based on the actually revised manuscript.

We would like to thank you again and hope that the revised manuscript can now be considered for publication.

Kind regards,
Maik Heistermann
(on behalf of the author team)

**1. Response to referee 1**

**1.1. CRNS Calibration**

**RC:** *[...] The only really major criticism I have concerns the calibration of the stationary cosmic ray neutron sensors (CRNS). It has been shown that highly variable systems like temperate forests require calibration for at least two points in time (one dry state and one wet state of the system, preferably), (see, e.g., Heidbüchel et al., 2016; Tan et al., 2020). Having the SoilNet sensors with continuous monitoring available, this can be done – however, the SoilNet does not cover the entire footprint of all the CRNS. Instead, one manual soil core sampling campaign was conducted to 'fill in the gaps' and get additional data on bulk density, lattice water content, etc. This allows for the proper calibration of the CRNS at (only) one point in time. I would expect a discussion of this deliberate choice to only use one point in time for calibration – in particular noticing the fact that in your case the calibrated CRNS seems to consistently overestimate soil moisture during the dry periods.*

AR: We thank you for pointing out this important issue. In the original manuscript, we had decided not to address it in depth since the prime focus of the paper is on the presentation of the data set and some of its fundamental properties. Based on your comment, though, we agree that it makes sense to discuss the different calibration options and the resulting implications for different use cases. We have hence added another paragraph to the subsection "Soil moisture estimation from stationary CRNS" in section 5 (ll. 459 ff.). As already elaborated in depth in the interactive discussion ( see this link), this additional paragraph addresses multiple issues (some of which you have also raised in section 1.2 ("Other specific comments"): the issue of the two-point calibration, the incorporation of the manual measurements (as compared to using only the SoilNet measurements), and the alternative of estimating one single $N_0$ value for all sensor locations (which could also a way to deal with the uncertainty that is caused by the fact that for many footprints, we do not have the recommended number of 18 reference measurements for calibration).

The additional paragraph reads as follows:

> Depending on the application context, prospective research should also explore other options to calibrate the relationship between observed neutron rates and soil moisture. Fig. 4b suggests the need for a second calibration date in September in order to avoid overestimation under drier conditions. Such a multi-point calibration ha already been suggested by [Iwema et al., 2015] and recommended specifically for forested environments by [Heidbüchel et al., 2016]. Furthermore, calibration in the above example was entirely based on SoilNet observations. That way, we increased the comparability between soil moisture estimated from CRNS and SoilNet. Including the manual measurements (section 3.8.2) would make soil moisture estimates more representative of the complete CRNS footprints in the periphery of the SoilNet area. Such estimates would then allow to cover the entire catchment, including the area south of the main road. In that case, however, a two-point calibration were not possible because the manual campaign was carried out just once. Alternatively, $N_0$ could be calibrated not individually for each sensor, but, simultaneously, as one single value for all sensor locations that have a sufficient number of reference measurements. This requires the careful consideration of hydrogen pools in vegetation and soil organic matter [Heistermann et al., 2021]. The transferability of such an $N_0$ value to locations with incomplete ground-truth coverage could be evaluated in cross-validation experiments. The estimation of a single $N_0$ value could also help to reduce the calibration uncertainty that is caused when each sensor is calibrated individually from a limited number of reference measurements. This is specifically relevant in the case of dense CRNS networks for which it is difficult to acquire, for each footprint, the amount of reference measurements typically recommended for calibration [Franz et al., 2012, Schrön et al., 2017].

At this point, we would, however, again like to emphasize that this paper is not about the study of soil moisture retrieval from CRNS, but about the presentation of the data set. While we consider the above paragraph as helpful guidance for users of the data set, we also think that further in-depth treatment of this issue is beyond the scope of this data paper.

**1.2. Other specific comments**

RC: *Abstract, Line 7: Do you mean 'watershed boundary'? 'Watershed' alone can also mean the catchment.*

AR: In fact, we have used the term watershed, in this context, in the sense of the "watershed boundary". As this is obviously ambiguous, we have replaced "watershed" by "catchment boundary".

RC: *Line 283-284, 307: In order to calibrate a CRNS properly it is recommended to independently measure*

*soil moisture twice (preferably under wet and under dry conditions, see, e.g. Heidbüchel et al., 2016; Tan et al., 2020). Also, it is recommended to have 18 different sampling locations within the footprint of a CRNS (see Franz et al., 2012). For many of the CRNS this requirement is not fulfilled. How do you justify this? At least it would be added value to provide a two-point calibration.*

AR: Please see our response to your first comment 1.1.

RC: *Line 311: What are these 18 locations? In Figure 2 there are more than 18 locations marked with 'manual sampling'.*

AR: Thank you for pointing out this possible misunderstanding. As you stated correctly in the next comment, there were 68 manual measurements of soil moisture profiles: 18 measurements with soil cores (gravimetric determination in the lab, see ll. 311-316), and 50 in-situ measurements with handheld FDR-sensors (see ll. 317-324). We had decided not to distinguish the two approaches in Fig. 2 for the sake of clarity. However, as the you explicitly requested the positions of the soil core sampling (next comment), we now distinguish the two approaches in Fig. 2, and also clarified, in the first paragraph of Sect. 3.8.2, the total number of manual samples.

RC: *Line 317: I guess that is where the extra 'manual samples' derive from. I would mark them differently in Figure 2. (I really want to know where you did the soil core sampling).*

AR: Please see our response to the previous comment.

RC: *Line 357: Why won't you tell us the exact number of plots in the grassland and in the shrubland areas?*

AR: We revised the manuscript accordingly to provide this information. Originally, we had intended to distinguish only two land cover types: forest and open land - which is why we assigned 15 plots to each type. However, during vegetation mapping and biomass sampling, it turned out that a few parts of the "open land" were already dominated by shrubs, which is why we differentiated into grassland (with 11 sample plots) and shrubland (with four sample plots). We added these numbers to the manuscript. The sentence now reads:

> For the quantification of above-ground dry biomass (Schmidt, 2021), we randomly selected 15 plots in the forest area and a total 15 plots in areas classified as grassland (11 plots) or shrubland (4 plots).

RC: *Line 409: What do you mean by 'complemented'?*

AR: By "complemented", we meant that we also contributed our improved land cover maps to the OSM community land use data. As this is not really relevant to the paper, we dropped "and complemented" from the revised version of the manuscript.

RC: *Line 445-448: You only used one date (and one condition) to calibrate N0, although you have SoilNet values for all kinds of soil moisture conditions. That is dangerous and potentially weakens the measurement performance of the CRNS. You can see in Fig. 4b that the CRNS overestimates soil moisture in dry conditions – this could have been avoided with another calibration performed when it's dry.*

AR: Please see our response to your first comment.

RC: *Also, what about the manual samples from the soil cores, did you use them at all for calibration? If so, how did you incorporate them? If not, why not? After all, SoilNet does not cover the entire footprint of all of the sensors. At least, I would like to see this discussed.*

AR: In our exemplary data analysis in section 5, we did not use the manual samples for calibration, but focused

on the SoilNet observations. We will explicitly discuss this choice as already elaborated in our response to comment 1.1.

**1.3. Technical corrections**

**RC:** *Abstract, Line 15: 'hillslope' instead of 'hill-slope'.*

AR: Corrected.

**RC:** *Abstract, Line 16: '...the retrieval OF soil water...'.*

AR: Corrected.

**RC:** *Line 39: 'Soon enough...'?*

AR: We replaced "Soon enough [...]" by

> "Soon after the feasibility of soil moisture observation with stationary CRNS had been demonstrated, a mobile CRNS sensor [...]"

**RC:** *Line 41: Schrön et al., which year?*

Thanks for spotting the broken reference. We fixed it according to the updated reference as follows:

Schrön, M. et al. (2021). Neutrons on rails: Transregional monitoring of soil moisture and snow water equivalent. Geophysical Research Letters, 48, e2021GL093924. doi:10.1029/2021GL093924

**RC:** *Line 350: '... of the forest IS rather homogeneous...'*

AR: Corrected.

**RC:** *Line 363: raspBerry.*

AR: Corrected.

**RC:** *Line 402: '...groundwater depth...' (no space between ground and water).*

AR: Corrected.

**RC:** *Line 403: North Rhine-Westphalia is the English word.*

AR: Corrected.

**RC:** *Line 438: 'Thenceforth' is quite archaic and literary. I have never seen it used in a scientific paper.*

We replaced "thenceforth" by "thereafter".

**2. Response to referee 2**

**Specific comments**

**RC:** *L19 – suggest "...limited, particularly when small scale variability is high."*

AR: Thanks for the suggestion. We shortened this sentence further so it now reads:

> "The spatial representativeness of conventional point-based soil moisture measurements is often limited by high small-scale variability [...]".

RC: *L23 – replace "confronted" with "limited by"*

AR: Implemented accordingly.

RC: *L32 – suggest "...100-150 m with a vertical depth..."*

AR: Implemented accordingly.

RC: *L39 – suggest replacing "soon enough" with "More recently"*

AR: As a response to a comment by referee 1, we suggest the following:

> "Soon after the feasibility of soil moisture observation with stationary CRNS had been demonstrated, a mobile CRNS sensor ("CRNS roving") was established [...]".

We prefer to keep "soon" instead of "more recently" because [Desilets et al., 2010] already published the first proof-of-concept for CRNS roving two years after [Zreda et al., 2008]. To clarify this, we added the reference "[Desilets et al., 2010]" after this sentence.

RC: *L83 – fix "between 10 and 170 m"*

AR: Fixed.

RC: *L105 – suggest "It is available via EUDAT (see Heistermann et al., 2021a)."*

AR: Implemented accordingly.

RC: *L113 – suggest "was required to;" then remove "to" from the start of each point*

AR: Implemented accordingly.

RC: *Figure 2 – would fit better after dot point rather than within*

AR: We agree. However, figure positioning will be entirely revised during type setting / copy editing with a two column format, so we'd prefer not to interfere at this point.

RC: *Table 1 – would fit better after the dot points rather than within*

AR: Please see our response to the previous comment.

RC: *L350 – 2nd sentence makes no sense*

AR: We corrected the error, so the sentence now reads:

> "The age structure of the forest is rather homogeneous as it was planted around 1946 after comprehensive clearances. Hence the spatial heterogeneity of the forest biomass is low in comparison to more structured and diverse forests [...]"

**RC:** *L412 and L413 – I am not sure exemplary is a good word to use. Do you just mean an example of data usage?*

**AR:** We agree that "exemplary" is not ideal. It was meant in terms of "non-exhaustive" / "by-example". We were not aware of the meaning of "exemplary" in the sense of "best practice". We hence applied the following changes:

- change the header of section 5 (l. 412) to "Examples of cross-scale soil moisture patterns in space and time"

- change l. 413 to "[...] we will provide selected examples to convey an idea of spatial and temporal soil moisture patterns as well as of differences between sensors at different horizontal and vertical scales."

**RC:** *Figure 6 caption – suggest "flights" rather than "rides"*

**AR:** Changed accordingly.

**RC:** *L524 change to "... B2HANDLE allow users to share ..."*

**AR:** Changed accordingly.

**References**

[Desilets et al., 2010] Desilets, D., Zreda, M., and Ferré, T. P. A. (2010). Nature's neutron probe: Land surface hydrology at an elusive scale with cosmic rays. *Water Resources Research*, 46:W11505.

[Franz et al., 2012] Franz, T. E., Zreda, M., Rosolem, R., and Ferre, T. (2012). Field validation of a cosmic-ray neutron sensor using a distributed sensor network. *Vadose Zone Journal*, 11(4):vzj2012.0046.

[Heidbüchel et al., 2016] Heidbüchel, I., Güntner, A., and Blume, T. (2016). Use of cosmic-ray neutron sensors for soil moisture monitoring in forests. *Hydrology and Earth System Sciences*, 20(3):1269–1288.

[Heistermann et al., 2021] Heistermann, M., Francke, T., Schrön, M., and Oswald, S. E. (2021). Spatio-temporal soil moisture retrieval at the catchment scale using a dense network of cosmic-ray neutron sensors. *Hydrology and Earth System Sciences*, 25(9):4807–4824.

[Iwema et al., 2015] Iwema, J., Rosolem, R., Baatz, R., Wagener, T., and Bogena, H. R. (2015). Investigating temporal field sampling strategies for site-specific calibration of three soil moisture–neutron intensity parameterisation methods. *Hydrology and Earth System Sciences*, 19(7):3203–3216.

[Schrön et al., 2017] Schrön, M., Köhli, M., Scheiffele, L., Iwema, J., Bogena, H. R., Lv, L., Martini, E., Baroni, G., Rosolem, R., Weimar, J., Mai, J., Cuntz, M., Rebmann, C., Oswald, S. E., Dietrich, P., Schmidt, U., and Zacharias, S. (2017). Improving calibration and validation of cosmic-ray neutron sensors in the light of spatial sensitivity. *Hydrology and Earth System Sciences*, 21(10):5009–5030.

[Zreda et al., 2008] Zreda, M., Desilets, D., Ferré, T. P. A., and Scott, R. L. (2008). Measuring soil moisture content non-invasively at intermediate spatial scale using cosmic-ray neutrons. *Geophysical Research Letters*, 35(21):L21402, 1–5.

---

## Author Response (AR2)

**Author Response to Editor Comments**

**Soil moisture observation in a forested headwater catchment: combining a dense cosmic-ray neutron sensor network with roving and hydrogravimetry at the TERENO site Wüstebach**

Maik Heistermann et al.
*Earth Syst. Sci. Data Discuss.,* `doi:10.5194/essd-2021-445`
* * *
RC: *Reviewer Comment*,     AR: *Author Response*,     ☐ Manuscript text

Dear Sibylle,

thank you for your feedback on the revised version of our manuscript, and for giving us the opportunity to address the issue of data uncertainty. Herewith, we would like to submit a revised version of the manuscript, and respond to your comments.

We agree that this issue is important, and, as you noted, already addressed in some parts of the manuscript (e.g. for the hydrogravimetry and lysimeter data). Following your suggestion, we added, where missing in section 3, brief statements on measurement uncertainties as well as on sources of uncertainties that govern soil moisture estimation. To that end, we mostly refer to previous publications in order to avoid inflating the manuscript unnecessarily.

However, we would like to emphasize that, beyond the uncertainty of the actual measurements, the uncertainty of soil moisture estimated from such measurements will be highly heterogeneous and elude any general statement; it will depend on the processing methods, on the target scale, and on the way different sensors are combined. For some of the data subsets, we are just starting to *understand* what governs the uncertainty of soil moisture estimates (e.g. for airborne roving). It is one of the motivations of this data paper to enable research on the quantification of uncertainties across sensors and scales, but it is obviously beyond its scope to comprehensively address the theoretical and technical dimensions of such an endeavour (and we also confident that this is not what you had in mind).

Given this lengthy "disclaimer", we tried to revise the manuscript along the spirit of your comment, and hope that the revised manuscript can now be considered for publication.

Kind regards,
Maik (on behalf of the author team)